# Voronoi Density Estimator for High-Dimensional Data: Computation, Compactification and Convergence

**Vladislav Polianskii***[1]    **Giovanni Luca Marchetti***[1]    **Alexander Kravberg**[1]    **Anastasiia Varava**

**Florian T. Pokorny**[1]    **Danica Kragic**[1]

[1]School of Electrical Engineering and Computer Science, Royal Institute of Technology (KTH), Stockholm, Sweden

## Abstract

The Voronoi Density Estimator (VDE) is an established density estimation technique that adapts to the local geometry of data. However, its applicability has been so far limited to problems in two and three dimensions. This is because Voronoi cells rapidly increase in complexity as dimensions grow, making the necessary explicit computations infeasible. We define a variant of the VDE deemed Compactified Voronoi Density Estimator (CVDE), suitable for higher dimensions. We propose computationally efficient algorithms for numerical approximation of the CVDE and formally prove convergence of the estimated density to the original one. We implement and empirically validate the CVDE through a comparison with the Kernel Density Estimator (KDE). Our results indicate that the CVDE outperforms the KDE on sound and image data.

## 1 INTRODUCTION

Given a discrete set of data sampled from an unknown probability distribution, the aim of density estimation is to recover the underlying Probability Density Function (PDF) (Diggle 2013; Scott 2015). Non-parametric methods achieve this by directly computing the PDF through a closed formula, avoiding the potentially expensive need of searching for optimal parameters.

One of the most common non-parametric density estimation techniques is the Kernel Density Estimator (KDE; Gramacki 2018). The resulting PDF is a convolution between a fixed kernel and the discrete distribution of samples. In case of the Gaussian kernel, this corresponds to a mixture density with a Gaussian distribution centered at each sample. Another

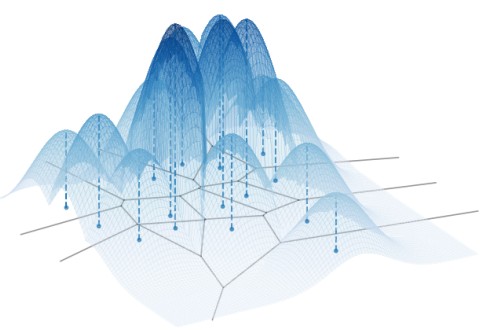

Figure 1: Graph of a density estimated by the CVDE, with the Voronoi tessellation underneath.

popular density estimator, more commonly used for visualization purposes is given by histograms (Freedman and Diaconis 1981), which depend on a prior tessellation of the ambient space (typically, a grid). The estimation is piecewise constant and is obtained by the number of samples falling in each cell normalised by its volume.

A common limitation of the aforementioned methods is a bias towards a fixed local geometry. Namely, estimates through KDE near a sample are governed by the level sets of the chosen kernel. In the Gaussian case, such level sets are ellipsoids of high estimated probability. Histograms suffer from an analogous bias towards the geometry of the cells of the tessellation (i.e., the bins of the histograms), on which the estimated PDF is constant. The issue of geometrical bias severely manifests when considering real-world high-dimensional data. Indeed, one cannot expect to approximate the rich local geometries of complex data with a simple fixed one. Both the estimators come with hyperparameters controlling the scale of the local geometries which require tuning. This amounts to the bandwidth for KDE and the diameter of the cells for histograms.

The *Voronoi Density Estimator* (VDE) has been suggested to tackle the challenges discussed above (Ord 1978). By con-

---

*Equal contribution.

*Accepted for the 38th Conference on Uncertainty in Artificial Intelligence* (UAI 2022).

sidering the Voronoi tessellation generated by data (Okabe et al. 2009), the estimated PDF is piece-wise constant on the cells and proportional to their inverse volume. The Voronoi tessellation adapts local polytopes so that each datapoint is equally likely to be the closest when sampling from the resulting PDF. This has enabled successful application of the VDE to geometrically articulated real-world distributions in lower dimensions (Duyckaerts, Godefroy, and Hauw 1994; Ebeling and Wiedenmann 1993; Vavilova et al. 2021).

The goal of the present work is to enable the VDE for high-dimensional scenarios. Although the VDE constitutes a promising candidate due to its local adaptivity, the following aspects have to be addressed:

**Computation**. The Voronoi cells are arbitrary convex polytopes and their volume is thus challenging to compute explicitly, which yields the necessity for fast approximate computations.

**Compactification**. Data is often concentrated around low-dimensional submanifolds, which makes most of the ambient space empty and several Voronoi cells unbounded, i.e. of infinite volume (see Figure 3). One still needs to produce a finite estimate on those cells, a process we refer to as 'compactification'.

We propose solutions to the problems above. First, we present efficient algorithmic procedures for volume computation and sampling from the estimated density. We formulate the cell volumes as integrals over a sphere, which can then be approximated by Monte Carlo methods. Furthermore, we propose a sampling procedure for the distribution estimated by the VDE. This consists in randomly traversing the Voronoi cells via a 'hit-and-run' Markov chain (Chen and Schmeiser 1996). The proposed algorithms are highly parallelizable, allowing efficient computations on the GPU.

In order to compactify the cells, we place a finite measure on each of them by means of a fixed kernel (typically, a Gaussian one), leading to an altered version of the VDE which we refer to as *Compactified Voronoi Density Estimator* (CVDE). Figure 1 shows an example of an estimate by the CVDE on a simple two-dimensional dataset. All the computational and sampling procedures naturally extend to the CVDE.

A further contribution of the present work is a theoretical proof of **convergence** for the CVDE. Assuming the original density has support in the whole ambient space, we show that the PDF estimated by the CVDE converges (with respect to an appropriate notion for random measures) to the ground-truth one as the number of datapoints increases. The convergence holds without any continuity assumptions on the ground-truth PDF nor on the kernel and does not require the kernel bandwidth to vanish asymptotically. This is in contrast with the convergence properties of the KDE. Due to the aforementioned local geometric bias of the KDE, the

bandwidth has to decrease at an appropriate rate in order to amend for the local influence of the kernel and guarantee convergence to the underlying distribution (Devroye and Wagner 1979; Jiang 2017).

Finally, we implement the CVDE in $C++$ and parallelize computations via the OpenCL framework. Our code, with a provided Python interface, is publicly available at `https://github.com/vlpolyansky/cvde`.

## 2 COMPACTIFIED VORONOI DENSITY ESTIMATOR

This section presents Voronoi cell compactification and Compactified Voronoi Density Estimator, CVDE. We begin by defining the Voronoi tessellations in a general setting (see Okabe et al. 2009 for a comprehensive treatment). Suppose that $(X, d)$ is a connected metric space and $P \subseteq X$ is a finite collection of distinct points referred to as *generators*.

**Definition 2.1.** The *Voronoi cell*[1] of $p \in P$ is defined as

$$C(p) = \{x \in X \mid \forall q \in P \; d(x, q) \geq d(x, p)\}. \quad (1)$$

The Voronoi cells intersect at the boundary and cover the ambient space $X$. The collection $\{C(p)\}_{p \in P}$ is called *Voronoi tessellation* generated by $P$. For a point $x \in X$ not on the boundary of any cell, we write $C(x)$ for the unique cell containing it. When $X = \mathbb{R}^n$ with Euclidean distance, the Voronoi cells are convex $n$-dimensional polytopes which are possibly unbounded.

Assume now that $X$ is equipped with a finite Borel measure denoted by Vol. An additional technical condition is that the boundaries of the Voronoi cells have vanishing measure.

**Definition 2.2.** The *Voronoi Density Estimator* (VDE) at a point $x \in X$ is defined almost everywhere as

$$\widetilde{f}(x) = \frac{1}{|P| \, \mathrm{Vol}(C(x))} \quad (2)$$

where $|\cdot|$ denotes cardinality.

The function $\widetilde{f}$ defines a locally constant PDF on $X$ and thus a probability measure $\widetilde{f}$ Vol. With respect to this distribution the cells are equally likely, and the restriction to each cell coincides with the normalisation of Vol.

We focus on the case where $X = \mathbb{R}^n$ equipped with Euclidean distance. One major issue for the choice of Vol is that the standard Lebesgue measure does not satisfy the finiteness requirement. A common solution in the literature is to restrict the measure to a fixed bounded region $A \subseteq \mathbb{R}^n$

---

[1]Sometimes referred to as *Dirichlet cell*.

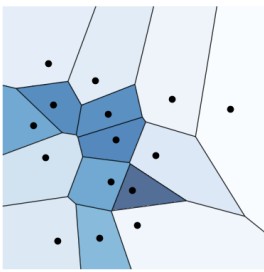

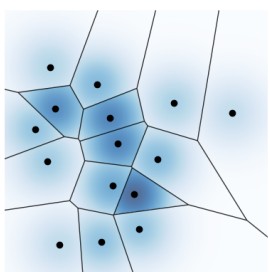

VDE with bounding square A          CVDE with Gaussian kernel

Figure 2: Comparison between VDE and CVDE for generators in the plane. A darker color represents higher estimated density.

containing $P$ (Moradi et al. 2019; Barr and Schoenberg 2010), which is equivalent to setting $X = A$ as the ambient space. However, this results in an often unsuitable solution for high-dimensional data. Under the manifold hypothesis (Fefferman, Mitter, and Narayanan 2016), data are concentrated around a submanifold with high codimension which implies that most of $\mathbb{R}^n$ falls outside the support. Moreover, the cells of the points lying at the boundary of the convex hull of data, which constitute the majority of cells for such submanifolds, are unbounded (see Figure 3). Estimating the density as uniform, after eventually intersecting with the bounded region $A$, becomes thus unreasonable and heavily relies on the a priori choice of $A$.

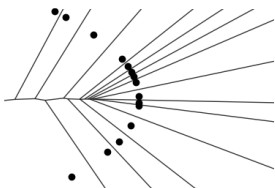

Figure 3: Voronoi tessellation for generators distributed on a submanifold (a parabola). In this case, all the Voronoi cells are unbounded and the VDE is strongly biased by the choice of the bounding region $A$.

We instead take a different route. The idea is to make the measure of each cell finite ('compactify') by considering a *local* distribution with mode at the corresponding generator in $P$. In general terms, we fix a positive kernel $K : \mathbb{R}^n \times \mathbb{R}^n \to \mathbb{R}_{\geq 0}$ which is at least integrable in the second variable and define the following:

**Definition 2.3.** The *Compactified Voronoi Density Estimator* (CVDE) at a point $x \in \mathbb{R}^n$ is defined almost everywhere as

$$f(x) = \frac{K(p, x)}{|P| \operatorname{Vol}_p(C(x))} \tag{3}$$

where $\operatorname{Vol}_p(C(x)) = \int_{C(x)} K(p, y) \, \mathrm{d}y$ and $p$ is the genera-

tor of $C(x)$ i.e., the generator $p \in P$ closest to $x$.

In practice, a commonly considered kernel is the Gaussian one

$$K(p, x) = e^{-\frac{\|p - x\|^2}{2h^2}} \tag{4}$$

where $h \in \mathbb{R}_{>0}$ is a hyperparameter referred to as 'bandwidth'. More generally, with abuse of notation a kernel can be constructed from an arbitrary integrable map $K \in L^1(\mathbb{R}^n)$:

$$K(p, x) = K\left(\frac{p - x}{h}\right). \tag{5}$$

Note that the VDE with a bounding region $A$ corresponds to the particular case of the CVDE with the characteristic function of $A$ as kernel i.e., $K(p, x) = \chi_A(x)$. Figure 2 shows a comparison between the VDE and the Gaussian CVDE on a simple two-dimensional dataset.

It is worth to briefly compare the CVDE to the Kernel Density Estimator (KDE). Recall that the KDE with kernel $K$ (which is assumed to integrate to 1 in the second variable) is given by $\frac{1}{|P|} \sum_p K(p, x)$. The kernel is aggregated over all the generators, which can possibly oversmooth the estimation. In contrast, the CVDE $f(x)$ involves $K$ evaluated at the closest generator alone. Furthermore, assume that all the cells have the same local volume i.e., $\operatorname{Vol}_p(C(p)) = 1$ for all $p \in P$, and that $K$ monotonically decreases with respect to the distance i.e., $K(p, x) \leq K(p', x)$ when $d(p, x) \geq d(p', x)$. Then the CVDE reduces to

$$f(x) = \frac{1}{|P|} \max_{p \in P} K(p, x) \tag{6}$$

which is a variant of the KDE where the sum gets replaced by a maximum. Such distributions are sometimes referred to as 'max-mixtures' (Olson and Agarwal 2013). An empirical comparison with KDE is presented in our experimental section (Section 6.4).

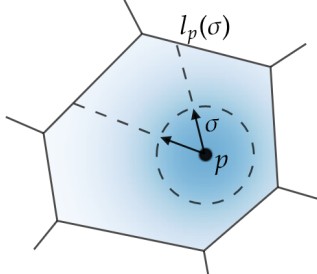

Figure 4: An illustration of the directional radius involved in volume estimation and sampling.

# 3 ALGORITHMIC PROCEDURES

The CVDE presents a number of computational challenges in high dimensions ($n \gg 3$) due to the increasing geometric complexity of Voronoi tessellations. We propose to deploy raycasting methods on polytopes which reduce the problem to one-dimensional subspaces. In the context of Voronoi tessellations raycasting has been considered to explore the boundaries of the cells in (Mitchell et al. 2018), which has led to a US Patent (Ebeida 2019), as well as in (Polianskii and Pokorny 2020). We utilize these techniques for volume computation and point sampling, and improve the time complexity through pre-computations and parallelization.

We first introduce an algebraic quantity necessary for the subsequent methods. Consider an arbitrary versor $\sigma$ and a point $z \in \mathbb{R}^n$. Define $l_z(\sigma)$ as the maximum $t$ such that $z + t\sigma$ is contained in $C(z)$, and $l_z(\sigma) = \infty$ if such $t$ does not exist. We refer to this value as a *directional radius*, originating at $z$ in the direction $\sigma$ (see Figure 4). The directional radius can be expressed via a closed and computable formula. Denote by $p$ the generator closest to $z$ and for $q \in P \setminus \{p\}$, set

$$l_z^q(\sigma) = \frac{\|q - z\|^2 - \|p - z\|^2}{2\langle \sigma, q - p \rangle}. \qquad (7)$$

As shown in (Polianskii and Pokorny 2019), the directional radius is given by

$$l_z(\sigma) = \min_{q \neq p, \, l_z^q(\sigma) \geq 0} l_z^q(\sigma) \qquad (8)$$

with $l_z(\sigma) = \infty$ if $l_z^q(\sigma)$ is negative for all $q$.

## 3.1 VOLUME ESTIMATION AND SAMPLING

We now present a way to efficiently compute the (local) volumes $\mathrm{Vol}_p$ via spherical integration. Such an approach to integration over high-dimensional Voronoi tessellations has been explored in the past by (Winovich et al. 2019) and (Polianskii and Pokorny 2019).

Assume that the kernel is as in Equation 5 for a continuous $K$. By a change of variables into spherical coordinates cen-

tered at $p$ and due to convexity of $C(p)$, the volumes can be rewritten as an integral over the unit sphere $\mathbb{S}^{n-1} \subseteq \mathbb{R}^n$:

$$\mathrm{Vol}_p = \int_{\mathbb{S}^{n-1}} \int_{[0, l_p(\sigma)]} K(t\sigma) t^{n-1} \mathrm{d}t \mathrm{d}\sigma \qquad (9)$$

where $l_p(\sigma)$ is the directional radius of the cell originating from its generator ($z = p$). The spherical integral can be computed via Monte Carlo approximation by sampling a finite set of versors $\Sigma_p \subseteq \mathbb{S}^{n-1}$ uniformly and estimating the empirical average

$$\frac{2\pi^{\frac{n}{2}}}{|\Sigma_p| \Gamma(\frac{n}{2})} \sum_{\sigma \in \Sigma_p} \int_{[0, l_p(\sigma)]} K(t\sigma) t^{n-1} \mathrm{d}t \qquad (10)$$

where $\Gamma$ denotes Euler's Gamma function. In the case of Gaussian kernel (Equation 4), by bringing the constant $\mathrm{Vol}(\mathbb{S}^{n-1}) = \frac{2\pi^{\frac{n}{2}}}{\Gamma(\frac{n}{2})}$ under the summation the summand simplifies to $(2\pi h^2)^{\frac{n}{2}} \overline{\gamma}\left(\frac{n}{2}, l_p(\sigma)\right)$, where $\overline{\gamma}$ denotes the regularized lower incomplete Gamma function $\overline{\gamma}(a, z) = \frac{1}{\Gamma(a)} \int_0^z t^{a-1} e^{-t} \mathrm{d}t$.

Next, we propose a sampling procedure for the CVDE which is a version of the *hit-and-run* sampling for distributions on higher-dimensional polytopes (Chen and Schmeiser 1996). It consists in first choosing a generator $p = z^{(0)} \in P$ uniformly. Then, one traverses the cell $C(p)$ by constructing a Markov chain $\{z^{(i)}\}$ in the following way. A random versor $\sigma^{(i+1)} \in \mathbb{S}^{n-1}$ is sampled uniformly and the next point $z^{(i+1)}$ is sampled from $\frac{1}{\mathrm{Vol}_p} K(p, \cdot)$ restricted to the segment $\{z^{(i)} + t\sigma^{(i+1)} \mid t \in [-l_{z^{(i)}}(-\sigma^{(i+1)}), \, l_{z^{(i)}}(\sigma^{(i+1)})]\}$. As shown by Chen and Schmeiser 1996, the Markov chain converges w.r.t. total variation distance to the underlying distribution $\frac{1}{\mathrm{Vol}_p} K(p, \cdot)$ over $C(p)$. In practice, one terminates the sampling process after a number $I$ of steps returning the last point $z^{(I)}$. Figure 5 shows an instance of hit-and-run on a simple two-dimensional dataset.

## 3.2 COMPUTATIONAL COMPLEXITY

The computational optimizations deserve a separate discussion. As seen from Equations 8 and 7, the natural way of estimating the directional radius $l_z(\sigma)$ for given $z \in \mathbb{R}^n$ and $\sigma \in \mathbb{S}^{n-1}$ would require $O(n|P|)$ numerical operations. This would bring the overall computational cost to $O(n \max_p |\Sigma_p| |P|^2)$ for the spherical integrals and to $O(n|P|I)$ for a sampling run with $I$ hit-and-run steps.

In order to optimize the algorithms, we first rewrite Equation 7 as

$$l_z^q(\sigma) = \frac{\langle q, q \rangle - \langle p, p \rangle - 2\langle z, q \rangle + 2\langle z, p \rangle}{2\langle \sigma, q \rangle - 2\langle \sigma, p \rangle}. \qquad (11)$$

In spherical integration, we deploy the same set of versors $\Sigma = \Sigma_p \subset \mathbb{S}^{n-1}$ for all the generators. This allows to pre-compute $\langle q, p \rangle$ and $\langle \sigma, p \rangle$ for all $p, q \in P, \sigma \in \Sigma$, achieving

**Algorithm 1** $\text{Vol}_p$ computation with Gaussian kernel

**Input:** $P \subset \mathbb{R}^n$ set of generators
$\quad \Sigma \subset \mathbb{S}^{n-1}$ set of versors
**Output:** $\text{Vol}_p$ for all $p \in P$
$\quad$ Compute $\langle q, p \rangle$ **for all** $(q, p) \in P \times P$
$\quad$ Compute $\langle \sigma, p \rangle$ **for all** $(\sigma, p) \in \Sigma \times P$
$\quad$ **for all** $p \in P$ **do**
$\qquad$ Initialize $\text{Vol}_p \leftarrow 0$
$\qquad$ **for all** $\sigma \in \Sigma$ **do**
$\qquad\quad$ Initialize $l_p(\sigma) \leftarrow \infty$
$\qquad\quad$ **for all** $q \in P \setminus \{p\}$ **do**
$\qquad\qquad l_p^q(\sigma) \leftarrow \frac{\langle q,q \rangle - 2\langle q,p \rangle + \langle p,p \rangle}{2\langle \sigma,q \rangle - 2\langle \sigma,p \rangle}$
$\qquad\qquad$ **if** $l_p^q(\sigma) > 0$ **then**
$\qquad\qquad\quad l_p(\sigma) \leftarrow \min\{l_p(\sigma), l_p^q(\sigma)\}$
$\qquad\qquad$ **end if**
$\qquad\quad$ **end for**
$\qquad\quad \text{Vol}_p \leftarrow \text{Vol}_p + |\Sigma|^{-1} \left(2\pi h^2\right)^{\frac{n}{2}} \overline{\gamma}\left(\frac{n}{2}, l_p(\sigma)\right)$
$\qquad$ **end for**
$\quad$ **end for**

---

a total computational complexity of $O(n|P|^2 + n|\Sigma||P| + |\Sigma||P|^2)$.

For the sampling procedure, we similarly fix a prior finite set $\Sigma$ of all available versors. This does not affect the convergence property of the hit-and-run Markov chain assuming $\Sigma$ linearly spans $\mathbb{R}^n$ (Bélisle, Romeijn, and Smith 1993). While $\langle \sigma, p \rangle$ and $\langle q, p \rangle$ can be pre-computed in $O(n|P|^2 + n|\Sigma||P|)$ time, the terms involving $z$ in Equation 11 require more care. To that end, the $i$-th step of the hit-and-run Markov chain is given by $z^{(i)} = z^{(i-1)} + t^{(i-1)}\sigma^{(i-1)}$ for appropriately sampled $t^{(i-1)}, \sigma^{(i-1)}$. The term $\langle z, p \rangle$ can then be updated inductively in $O(1)$ as $\langle z^{(i)}, p \rangle = \langle z^{(i-1)}, p \rangle + t^{(i-1)} \langle \sigma^{(i-1)}, p \rangle$. Summing up, the cost of a hit-and-run Markov chain run reduces to $O((|\Sigma| + |P|)I)$, which does not depend on the space dimensionality $n$ multiplicatively.

Algorithms 1 and 2 provide a more detailed description of volume computation and point sampling via the hit-and-run procedure respectively, including the discussed optimizations. Note that the loops in both algorithms are independent and involve elementary algebraic operations. This allows to utilize GPU capabilities, which also significantly boosts the computation performance.

# 4 THEORETHICAL PROPERTIES

## 4.1 CONVERGENCE

We now discuss the convergence of the CVDE when the set $P$ of generators is sampled from an underlying distribution. Suppose thus that there is an absolutely continuous probability measure $\mathbb{P} = \rho \mathrm{d}x$ on $\mathbb{R}^n$ defined by a density $\rho \in L^1(\mathbb{R}^n)$. When $P$ is sampled from $\mathbb{P}$ the CVDE can be considered as (the density of) a random probability measure.

**Algorithm 2** CVDE sampling

**Input:** $P \subset \mathbb{R}^n$ set of generators
$\quad \Sigma \subset \mathbb{S}^{n-1}$ set of versors
$\quad m$ desired number of samples
$\quad I$ number of hit-and-run steps
**Output:** $Z = Z^{(I)} \subset \mathbb{R}^n$ samples from CVDE
$\quad$ Initialize $Z^{(0)} \sim \text{Uni}^m(P)$
$\quad$ Compute $\langle p, p \rangle$ **for all** $p \in P$
$\quad$ Compute $\langle z, p \rangle$ **for all** $(z, p) \in Z^{(0)} \times P$
$\quad$ Compute $\langle \sigma, p \rangle$ **for all** $(\sigma, p) \in \Sigma \times P$
$\quad$ **for** $i = 1$ **to** $I$ **do**
$\qquad$ **for all** $z \in Z^{(i-1)}$ **do**
$\qquad\quad \sigma \leftarrow \text{Uni}(\Sigma), p \leftarrow z^{(0)}$
$\qquad\quad$ Initialize $l_z(-\sigma) \leftarrow \infty, l_z(\sigma) \leftarrow \infty$
$\qquad\quad$ **for all** $q \in P \setminus \{p\}$ **do**
$\qquad\qquad l_z^q(\sigma) \leftarrow \frac{\langle q,q \rangle - \langle p,p \rangle - 2\langle z,q \rangle + 2\langle z,p \rangle}{2\langle \sigma,q \rangle - 2\langle \sigma,p \rangle}$
$\qquad\qquad$ **if** $l_z^q(\sigma) > 0$ **then**
$\qquad\qquad\quad l_z(\sigma) \leftarrow \min\{l_z(\sigma), l_z^q(\sigma)\}$
$\qquad\qquad$ **else**
$\qquad\qquad\quad l_z(-\sigma) \leftarrow \min\{l_z(-\sigma), -l_z^q(\sigma)\}$
$\qquad\qquad$ **end if**
$\qquad\quad$ **end for**
$\qquad\quad$ Sample $t \in [-l_z(-\sigma), \ l_z(\sigma)]$
$\qquad\quad$ Add $z + t\sigma$ to $Z^{(i)}$
$\qquad\quad$ Update $\langle z, p \rangle \leftarrow \langle z, p \rangle + t \langle \sigma, p \rangle$ **for all** $p \in P$
$\qquad$ **end for**
$\quad$ **end for**

---

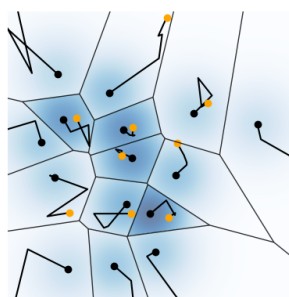

Figure 5: An illustration of the hit-and-run sampling procedure, with a trajectory of length $I = 4$ for each generator. The sampled points are displayed in orange.

We denote by $\mathbb{P}_m$ this random measure when the number of generators is $m$ i.e., $\mathbb{P}_m = f\mathrm{d}x$ for $P \sim \rho^m$.

The following is our main theoretical result. It guarantees that $\mathbb{P}_m$ converges to $\mathbb{P}$ with respect to a canonical notion of convergence for random measures, assuming $\rho$ has full support.

**Theorem 4.1.** *Suppose that $\rho$ has support in the whole $\mathbb{R}^n$. For any $K \in L^1(\mathbb{R}^n \times \mathbb{R}^n)$ the sequence of random probability measures $\mathbb{P}_m$ converges to $\mathbb{P}$ in distribution w.r.t. $x$ and in probability w.r.t. $P$. Namely, for any measurable set $E \subseteq \mathbb{R}^n$ the sequence $\mathbb{P}_m(E)$ of random variables*

*converges in probability to the constant* $\mathbb{P}(E)$.

*Proof.* We outline here an idea of the proof and refer to the Appendix for full details. For a measurable set $E$, $\mathbb{P}_m(E)$ is equal to

$$\frac{1}{m}|P \cap E| + residue \qquad (12)$$

where the residue bounded by (twice) the relative number $R$ of generators whose Voronoi cell intersects the boundary $\partial E$ of $E$. The variable $\frac{1}{m}|P \cap E|$ tends to $\mathbb{P}(E)$ in probability by the law of large numbers.

We then proceed to show that the boundary term $R$ tends to 0 in probability. To this end, we first prove that the diameters of the Voronoi cells intersecting $E$ tend uniformly to 0, which in turn requires a preliminary result constraining such cells in a neighbour of $E$ (which is assumed to be bounded). Given that, we conclude that $R$ tends to $\mathbb{P}(\partial E)$ by the law of large numbers. By the Portmanteau Lemma (Van der Vaart 2000), we can assume that $\mathbb{P}(\partial E) = 0$ (and that $E$ is bounded), which concludes the proof. $\square$

Note that the above results holds for any (integrable) kernel, thus even for discontinuous ones. The kernel is fixed, and there is no need for an eventual bandwidth (Equation 5) to vanish asymptotically. This is in contrast with KDE, which requires $h$ to tend to 0 at an appropriate rate in order to obtain convergence to $\rho$ (Devroye and Wagner 1979; Jiang 2017). This is because of the local geometric bias inherent to the KDE, as discussed in Section 1. In order to obtain convergence, such bias has to be amended with a vanishing bandwidth that annihilates the local geometry of the kernel.

We remark that the assumption on the support of $\rho$ in Theorem 4.1 is satisfied in the presence of noise, which is realistic in practical scenarios. Assuming that data exhibit, say, Gaussian noise, the actual underlying distribution is of full support even when the ideal one is concentrated on a submanifold of $\mathbb{R}^n$.

## 4.2 BANDWIDTH ASYMPTOTICS

Consider a kernel in the form of Equation 5. The asymptotics with respect to $h$ (with fixed set of generators $P$) can be easily deduced:

**Proposition 4.2.** *For a continuous $K : \mathbb{R}^n \to \mathbb{R}_{\geq 0}$, the following hold:*

*(i) As $h$ tends to $0$, $f$ converges in distribution to the empirical measure $\frac{1}{|P|} \sum_{p \in P} \delta_p$, where $delta_p$ denotes the Dirac's delta centered in $p$ i.e., the probability measure concentrated in the singleton $\{p\}$.*

*(ii) Consider the restriction of the kernel to a bounded region $A$ (i.e., its product with $\chi_A$). As $h$ tends to $+\infty$, $f$ converges in distribution to the VDE $\widetilde{f}$.*

*Proof.* For the first statement, note that $\frac{1}{h^n}K(\frac{x}{h})$ tends to $K(0)\delta_0$ in distribution by the general theory of approximators of unity. Since $\lim_{h \to 0} \mathrm{Vol}_p(C(x)) = K(0)$ as well for every $p$, the claim follows from the definition of the CVDE (Equation 3). As for the second part, observe that $K(x, p)$ tends to $K(0)$ by continuity of $K$ and thus $f(x)$ tends to $\widetilde{f}(x)$ for almost every $x$. To conclude, pointwise convergence of PDFs implies convergence in distribution (Scheffé's Lemma). $\square$

The asymptotics for small bandwidth are the same as for the KDE. For bandwidth tending to infinity, however, the KDE tends to the uniform distribution over $A$, while the CVDE still gives reasonable estimates in the form of its non-compactified version.

## 5 RELATED WORK

**Non-parametric Density Estimation**. The first traces of systematic density estimation date back to the introduction of histograms (Pearson 1894). Those have been subsequently considered with a variety of cell geometries such as rectangles, triangles (Scott 1988) and hexagons (Carr, Olsen, and White 1992). The choice of geometry constitutes the main source of bias for the histogram-based density estimator.

Arguably, the most popular density estimator is the KDE, first discussed by Rosenblatt 1956 and Parzen 1962. Numerous extensions have followed, for example, to the multivariate case (Izenman 1991; Dehnad 1987), bandwidth selection methods (Marron 1987; Wand, Jones, et al. 1994) and algorithms for adaptive bandwidths (B. Wang and X. Wang 2007; Walt and Barnard 2017). The latter aim to partially amend for the local geometric bias of the KDE, which is in line with the present work. However, adapting the bandwidth alone provides a partial solution since it enables different scales of the same local geometry. Among applications, the KDE has been deployed to estimate traffic incidents (Xie and Yan 2008), archeological data (Baxter, Beardah, and Wright 1997) and wind speed (Bo et al. 2017) to name a few.

**VDE and its Applications**. The VDE has been originally introduced by Ord 1978 under the name 'ideal estimator' because of its local geometric adaptivity. Subsequent works have discussed regularisation (Moradi et al. 2019) and lower-dimensional aspects (Barr and Schoenberg 2010). The VDE has seen a applications to a variety of real-world densities such as neurons in the brain (Duyckaerts, Godefroy, and Hauw 1994), photons (Ebeling and Wiedenmann 1993) and stars in a galaxy (Vavilova et al. 2021). Although promising, the VDE has been previously limited to low-dimensional problems.

**Theoretical Convergence**. Convergence of the VDE

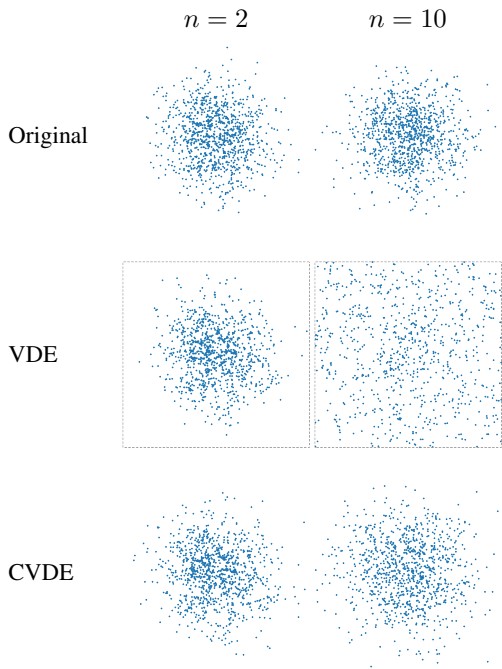

$n = 2$  $n = 10$

Original

VDE

CVDE

Figure 6: Visual comparison between samples from the CVDE and the VDE estimating an $n$-dimensional Gaussian for $n = 2, 10$. In the 10-dimensional case, points are projected onto a plane. In high dimensions, the VDE appears as biased towards a uniform distribution. This is because of abundance of unbounded cells, over which the estimated density is constant.

has been previously considered in the literature, usually in the language of Poisson point processes. For uniform underlying distribution, pointwise convergence of the averaged estimated density (i.e., unbiasedness: $\lim_{m \to \infty} \mathbb{E}_{P \sim \rho^m}[\widetilde{f}(x)] = \rho(x)$ for almost all $x$) has been proven by Last 2010. For non-uniform distributions, the same convergence has been shown by Moradi et al. 2019 with strong continuity assumptions on the density, which allows a reduction to the uniform case. Our theoretical result is based on a different, non-averaged notion of convergence and holds for the more general CVDE with no continuity assumptions.

# 6   EXPERIMENTS

## 6.1   DATASET DESCRIPTION

In our experiments, we evaluate the CVDE on datasets of different nature: simple *synthetic distributions* of Gaussian type, *image data* in pixel-space, and *sound data* in a frequency space. The datasets we deploy are the following:

**Gaussians and Gaussian Mixtures:** for synthetic experiments we generate two types of datasets, each contain-

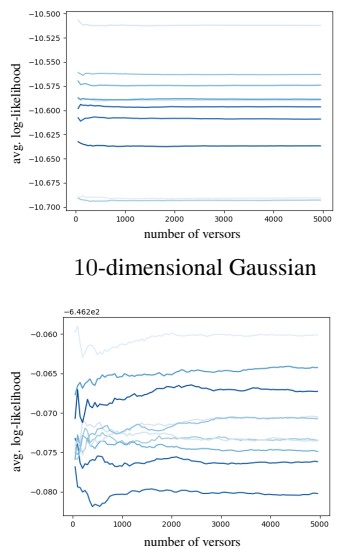

10-dimensional Gaussian

MNIST

Figure 7: Stabilisation of the Monte Carlo spherical integral. The plots display the average log-likelihood of the estimated density on the training set as the number of sampled versors increases. For each of the 2 datasets, 10 experimental runs are shown.

ing 1000 training and 1000 test points. The first one consists of samples from an $n$-dimensional standard Gaussian distribution. The second one is sampled from a Gaussian mixture density $\rho = \frac{1}{2}(\rho_1 + \rho_2)$. Here, $\rho_1, \rho_2$ are Gaussian distributions with means $\mu_1 = (-0.5, 0, \cdots, 0)$, $\mu_2 = (0.5, 0, \cdots, 0)$ and standard deviations $\sigma_1 = 0.1$, $\sigma_2 = 100$ respectively.

**MNIST** (Deng 2012): the dataset consists of $28 \times 28$ grayscale images of handwritten digits which are normalised in order to lie in $[0, 1]^{28 \times 28}$. For each experimental run, we sample half of the 60000 training datapoints in order to evaluate the variance of the estimation. The test set size is 10000.

**Anuran Calls** (Dua and Graff 2017): the datasets consists of 7195 calls from 10 species of frogs which are represented by 21 normalised mel-frequency cepstral coefficients in $[0, 1]^{21}$. We retain 10% of data for testing and again sample half of the training data at each experimental run.

## 6.2   COMPARISON WITH VDE

In this section, we evaluate empirically the necessity of compactification for high-dimensional data. To this end, we visually compare samples from the CVDE (with Gaussian kernel) and from the the VDE. The VDE is implemented with a bounding hypercube $A = [-\frac{7}{2}, \frac{7}{2}]^n$ as described in Section 2.

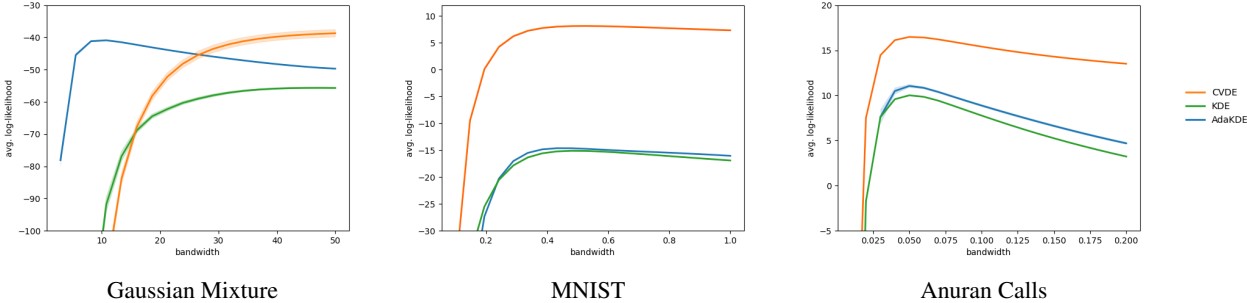

| | | |
|---|---|---|
| Gaussian Mixture | MNIST | Anuran Calls |

Figure 8: Empirical comparisons between the CVDE , the KDE and the KDE with adaptive bandwidth ( AdaKDE ). The plots display the average log-likelihood over the test set as the bandwidth varies. The shadowed region represents standard deviation (with respect to sampling of the dataset) on 5 experimental runs.

We consider the Gaussian dataset in $n = 2$ and $n = 10$ dimensions. For both the estimators, 1000 points are sampled via hit-and-run (with trajectories of length $I = 1000$) from the estimated density. The bandwidth for the CVDE is chosen following Scott's rule (Scott 2015) and amounts to $h = 0.33$ in two dimensions and to $0.66$ in ten dimensions.

The results are presented in Figure 6. In two dimensions, both the estimators produce samples that are visually close to the ground-truth distribution. However, in ten dimensions the sampling quality of VDE drastically decreases, while the CVDE still produces a satisfactory result. In the provided examples, more than $85\%$ of points sampled from the VDE belong to the Voronoi cells intersecting the boundary of $A$. Since the VDE is uniform within each cell, the estimation and the consequent sampling is biased by the choice of the bounding region $A$, especially in high dimensions.

## 6.3 CONVERGENCE OF THE SPHERICAL INTEGRAL

We now empirically estimate the amount of Monte Carlo samples required for spherical integration (Equation 10). To this end, we visualize how the approximation for the volumes in the CVDE (with Gaussian kernel) changes as the number $|\Sigma|$ of versors increases. We consider two datasets: the 10-dimensional Gaussian one and MNIST. Each plot in Figure 7 displays 10 curves, each corresponding to one experimental run. What is shown is the average log-likelihood of the estimated density on the training set, which correponds up to an additive constant to the average negative logarithmic volume $-\frac{1}{|P|}\sum_{p \in |P|} \log \mathrm{Vol}_p(C(p))$ of the Voronoi cells. The bandwidth is again chosen according to Scott's rule for the Gaussian dataset while it is set to 1 for MNIST. Evidently, all the curves are stable at $|\Sigma| = 5000$ sampled versors, which we fix as a parameter in later experiments.

## 6.4 COMPARISON WITH KDE

We now compare the CVDE with the KDE (both with Gaussian kernel) on the synthetic and real-world data described in Section 6.1. However, the distribution of high-dimensional real-world data is too sparse in the original ambient space to allow for a meaningful comparison. We consequently pre-process the MNIST and the Anuran Calls datasets via Principal Component Analysis (PCA) and orthogonally project them to the 10-dimensional subspace with largest variance. We set the dimension of the synthetic Gaussian mixture to 10 as well.

We compare the CVDE with the standard KDE as well as the KDE with local, adaptive bandwidths (AdaKDE) described in B. Wang and X. Wang 2007. In the AdaKDE the bandwidth $h_p$ depends on $p \in P$ and is smaller when data is denser around $p$. Specifically, denote by $\hat{f}(p)$ the standard KDE estimate with a global bandwidth $h$. Then $h_p = h\lambda_p$ where $\lambda_p = (g/\hat{f}(p))^{\frac{1}{2}}$ and $g = \prod_{q \in P} \hat{f}(q)^{\frac{1}{|P|}}$.

We score the estimators via the average log-likelihood on a test set i.e., $P_{\text{test}}$ i.e., $\frac{1}{|P_{\text{test}}|}\sum_{p \in P_{\text{text}}} \log f(p)$. Such score measures the adherence of the estimated density to the ground-truth one and penalizes overfitting thanks to the deployment of the test set.

The results are displayed in Figure 8 with the bandwidth varying for all the estimators. For AdaKDE we vary the global bandwidth for $\hat{f}$. Sampling of training and test data is repeated for 5 experimental runs, from which mean and standard deviation of the score are displayed.

When each estimator is considered with its best bandwidth, the CVDE outperforms the baselines. This shows that the local geometric adaptivity of the CVDE leads to density estimates that are closer to the ground-truth distribution. Moreover, the CVDE displays remarkably better scores as the bandwidth increases. This is consistent with the discussion in Section 4.2 as the CVDE has more informative

asymptotics than the KDE for large $h$. On the real-world datasets (MNIST and Anuran Calls), the adaptive bandwidth does not drastically improve the performance of KDE. On the synthetic data, the AdaKDE is instead competitive with the CVDE. This indicates that the local adaptivity of the AdaKDE is enough to capture simple densities such as a Gaussian mixture. However, for more complex distributions the AdaKDE still suffers from the bias due to the Gaussian kernel (albeit with a local bandwidth) as mentioned in Section 5. The CVDE instead effectively adapts to the local geometry of data via the Voronoi tessellation.

## 7 CONCLUSIONS AND FUTURE WORK

In this work, we defined an extension of the Voronoi Density Estimator suitable for high-dimensional data, providing efficient methods for approximate computation and sampling. Additionally, we proved convergence to the underlying data density.

A promising line of future research lies in exploring both theory and applications of the VDE and CVDE to metric spaces beyond the Euclidean one, in particular higher-dimensional Riemannian manifolds. Spheres, for example, naturally appear in the context of normalised data, while complex projective spaces of arbitrary dimension arise as Kendall shape spaces on the plane (Mardia and Jupp 2009).

## 8 ACKNOWLEDGEMENTS

This work was supported by the Swedish Research Council, the Knut and Alice Wallenberg Foundation and the European Research Council (ERC-BIRD-884807).

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
