# OpenReview forum: "Voronoi Density Estimator for High-Dimensional Data: Computation, Compactification and Convergence"
_auai.org/UAI/2022/Conference — UAI 2022 Poster_

### Official Review · Reviewer_z7n3 · 2022-04-07

**Q2(1) Originality/Novelty:** 3
**Q2(2) Significance/Impact:** 2
**Q2(3) Correctness/Technical Quality:** 3
**Q2(6) Clarity Of Writing:** 4
**Q6 Overall Score:** 7
**Q8 Confidence In Your Score:** 3

**Q1 Summary And Contributions:**

The authors propose an enhanced version of the VDE which is both computationally feasible and suited to data categories with many unbounded Voronoi cells (which are likely to occur when the data are concentrated on lower-dimensional submanifolds of the input space). They show that their proposed method asymptotically converges to the correct density, and provide empirical evidence that it also performs well for realistic data sets.

**Q2 Assessment Of The Paper:**

More detailed information regarding each of these aspects is given below:

**Q2(4) Quality Of Experiments (Optional):**

2: Fair: The experimental evaluation is weak: important baselines are missing, or the results do not adequately support the main claims.

**Q2(5) Reproducibility:**

4: Excellent: Key resources (e.g., proofs, code, data) are available and key details (e.g., proof sketches, experimental setup) are comprehensively described for competent researchers to confidently and easily reproduce the main results.

**Q3 Main Strengths:**

The authors carefully lay out both the strengths and the weaknesses of Voronoi density estimation, and convincingly demonstrate that their modification overcomes at least some of the weaknesses while maintaining the strengths of VDE. I did not check their proofs in detail, but they are credible and seem solid. The proposed computational approach, based on Monte Carlo approximation with raycasted samples is powerful (though not novel in the current submission). Finally, the empirical comparisons are careful and informative, though not entirely convincing for reasons I describe below.

**Q4 Main Weakness:**

I am concerned about two weaknesses, of which the more debatable one is philosophical: is density estimation (in any form) a useful pursuit for high-dimensional spaces? It is well known that our intuitions in low dimensions do not generalize well - and perhaps such intuitions are the main reason to attempt high-dimensional density estimation. In practice, any high-dimensional space is necessarily dominated by empty space for realistic data-set sizes, and density estimators do not seem like a natural way to extend the impact of populated regions to emptiness. This conclusion is given credence by the fact that the current submission (like much recent work on high-dimensional density estimation) relies on dimension reduction as soon as real-world data is encountered. My other concern is that the competitive baselines may not be the strongest candidates for evaluating the proposed method, as discussed in my detailed comments.

**Q5 Detailed Comments To The Authors:**

The paper is carefully written and well organized. There are, however, a few issues which bother me:
- The authors state that 'one cannot expect to approximate the rich local geometries of complex data with a simple fixed one.' I disagree: if the local 'kernels' are sufficiently adaptive, it is not clear that their simplicity is more problematic in high-dimensional spaces compared to low dimensions (where simple kernels clearly are sufficient to approximate many real-world distributions).
- When comparing the CVDE with KDE, the authors assume that the kernel K is fixed for all generators. However, there are many proposals for spatially variable K in the literature, and in their empirical comparison a KDE with local, adaptive bandwidths is indeed employed. Thus, the CVDE - KDE comparison comes across as a strawman argument.
- Similarly, adaptive bandwidth methods should be mentioned when related work is discussed. In fact, a stronger baseline for the empirical comparison is probably achievable by combining something like AdaKDE with a more powerful way to select the local kernel parameters - see, for example, Leiva-Murillo, J.M. and Artés-Rodríguez, A., 2012. Algorithms for maximum-likelihood bandwidth selection in kernel density estimators. Pattern Recognition Letters, 33(13), pp.1717-1724 and Van der Walt, C.M. and Barnard, E., 2017. Variable kernel density estimation in high-dimensional feature spaces. In Thirty-first AAAI conference on artificial intelligence.

**Q7 Justification For Your Score:**

Although I am personally not convinced that high-dimensional density estimation is a viable pursuit, the current submission certainly progresses that state of the art in that domain.

**Q9 Complying With Reviewing Instructions:**

1: Yes.

---

### Official Review · Reviewer_fenV · 2022-04-10

**Q2(1) Originality/Novelty:** 3
**Q2(2) Significance/Impact:** 2
**Q2(3) Correctness/Technical Quality:** 3
**Q2(6) Clarity Of Writing:** 2
**Q6 Overall Score:** 6
**Q8 Confidence In Your Score:** 3

**Q1 Summary And Contributions:**

This paper proposed a new non-parametric density estimator, which combines kernel density estimation (KDE) with Voronoi density estimation. The authors provided an efficient sampling procedure to estimate the density, whose complexity scales linearly with the dimensionality. The authors proved its convergence to the underlying true density.

**Q2 Assessment Of The Paper:**

More detailed information regarding each of these aspects is given below:

**Q2(4) Quality Of Experiments (Optional):**

2: Fair: The experimental evaluation is weak: important baselines are missing, or the results do not adequately support the main claims.

**Q2(5) Reproducibility:**

4: Excellent: Key resources (e.g., proofs, code, data) are available and key details (e.g., proof sketches, experimental setup) are comprehensively described for competent researchers to confidently and easily reproduce the main results.

**Q3 Main Strengths:**

The proposed CVDE density estimation in eq.(3) is **simple** and overcomes some challenges of KDE and VDE. I am a bit surprised that it has not been used before. This is a strength of this work.

The authors gave its basic idea, a computational algorithm with linear complexity, the convergence proof, and simple numerical experiments. Overall the proposed method is **completely** presented with good clarity.

**Q4 Main Weakness:**

The authors tested the proposed CVDE estimator on real high dimensional data including MNIST and Anuran Calls, showing it is better than the vanilla KDE and adaptive bandwidth KDE (AdaKDE). The experiments can be more comprehensive, including a set of baseline methods such as the Gaussian mixture model. Why VDE is not included in the experiments in Figure 7? This section can be better organized by using the same set of baseline methods in different experiments.

In the experiments, there should be a comparison of computational efficiency, which is missing. Obviously, the proposed method is more expensive than the KDE variants. It is good to give an intuitive idea of how much computational overhead is needed to achieve the good performance of CVDE.

**Q5 Detailed Comments To The Authors:**

Section 3. In the beginning, introduce the overall computational framework, and what high complexity computation is needed. In the current form, the "directional radius" related contents are not well-motivated, and one may wonder why it is needed.

Proposition 4.2. The statement reads a bit informal including the proof. It can be better stated in terms of (in)equalities.

Figure 5. can we put KDE into the figures? and what does it look like?

Can the proposed CVDE method lead to parametric generalizations?


****** After rebuttal ******

The authors provided reasonable explanations on why some baselines are missing in the experiments, and why the run time comparison is not included. I had another look at the paper after reading the author's response. While the rebuttal does not affect negatively my score, I still believe the experimental section has space for some improvement.  As the authors proposed a simple modification of some classical approaches, a thorough empirical evaluation is necessary. At the bottom line, I suggest the authors include what they have explained here in the next/final version.

**Q7 Justification For Your Score:**

Overall, this contribution is useful for extending KDE into real high-dimensional data and is well presented. As far as I can see, the weakness mentioned above is not fatal but rather for improving the experimental evaluation and presentation of the experiments section.


**Q9 Complying With Reviewing Instructions:**

1: Yes.

---

### Official Review · Reviewer_665k · 2022-04-19

**Q2(1) Originality/Novelty:** 3
**Q2(2) Significance/Impact:** 3
**Q2(3) Correctness/Technical Quality:** 3
**Q2(6) Clarity Of Writing:** 4
**Q6 Overall Score:** 7
**Q8 Confidence In Your Score:** 3

**Q1 Summary And Contributions:**

Voronoi density estimators (VDEs) are unsuitable for high dimensional data (>3), and thus while theoretically hold properties that make them preferable to say KDEs, they are not practical. . In order to get around this problem the authors suggest an approximation Compactified Voronoi Density Estimator (CVDE) including algorithms to i) volume computation ii) sampling the density and theoretical results on i) algorithmic efficiency ii) convergence.

**Q2 Assessment Of The Paper:**

More detailed information regarding each of these aspects is given below:

**Q2(4) Quality Of Experiments (Optional):**

2: Fair: The experimental evaluation is weak: important baselines are missing, or the results do not adequately support the main claims.

**Q2(5) Reproducibility:**

3: Good: Key resources (e.g., proofs, code, data) are available and key details (e.g., proofs, experimental setup) are sufficiently well-described for competent researchers to confidently reproduce the main results.

**Q3 Main Strengths:**

Making VDEs able applicable to higher dimension is a significant achievement, with several novel ideas in the paper contributing to this:
- Compactified VDEs are introduced by means of introducing a fixed kernel in each cell so that they have finite measure (this is important as data on low-dimensional submanifolds would otherwise lead to unbounded cells)
- CVDE are still expensive to compute, thus another contribution is in how to do this more efficiently (computing local volumes by spherical integration using monte carlo approximation, and using hit-run Markov chains to sample the estimated density) and clearly written algorithms for these as well as discussion of their efficiency'
- convergence proof (to true density)

**Q4 Main Weakness:**

The motivation to perform VDE in high dimensions is stated to be their expected superior performance vs KDEs yet the experimental results could be more expansive to support this. In particular for this work to be very impactful it would have to be, as authors claim, that "[for KDEs] The issue of geometrical bias severely manifests when considering real-world highdimensional data.", yet in the experimental section it is stated that "However, the distribution of high-dimensional real-world data is too sparse in the original ambient space to allow for a meaningful comparison. [Thus PCA is performed to reduce dimensions]". Perhaps I'm misinterpreting this but the statements seem slightly contradictory. For high dimensional VDE to be truly impactful in practice, would one hope that such preprocessing wasn't necessary to show the benefits. Maybe this would be the case for some different data?

**Q5 Detailed Comments To The Authors:**

Overall a pleasure to read and easy to understand. As said above, my only reservation is whether a true practical benefit over KDEs could be illustrated more expansively. Few minor comments:
- it would be interesting to get some concrete idea about the significant boost by parallelization/GPUs (in practical sense)
- in particular, even with above, computation must still be more expensive that for KDE and while perhaps obvious this should still be considered more clearly (but how much difference does this make in different practical applications, and considering above point about GPUs, is not given enough thought)
- visual comparison of in Fig 5 is fine but apart from the VDE (n=10), it is hard to really say much about original vs. CVDE (n=10) is there any quantitative way you could compare them
- also where is KDE in fig 5?

**Q7 Justification For Your Score:**

Paper makes a significant novel contribution, with likely a big impact in its own subfield with extensions possible e.g. to non-Euclidean spaces and algorithmic improvements. Paper is written with exemplary clarity.

**Q9 Complying With Reviewing Instructions:**

1: Yes.

---

### Official Review · Reviewer_7DCy · 2022-04-20

**Q2(1) Originality/Novelty:** 3
**Q2(2) Significance/Impact:** 2
**Q2(3) Correctness/Technical Quality:** 3
**Q2(6) Clarity Of Writing:** 4
**Q6 Overall Score:** 6
**Q8 Confidence In Your Score:** 3

**Q1 Summary And Contributions:**

The authors propose a density estimation technique which is based on the Voronoi Density Estimator (VDE), but also mainly suitable for higher-dimensional data, called the Compactified VDE (CVDE). The method learns a kernel density estimation for each Voronoi cell, where as a default the Gaussian kernel is used. The distribution is learned via a hit-and-run sampling scheme. The method is evaluated on synthetic data, MNIST and a sound data set against two competitors.

**Q2 Assessment Of The Paper:**

More detailed information regarding each of these aspects is given below:

**Q2(4) Quality Of Experiments (Optional):**

3: Good: The experimental evaluation is adequate, and the results convincingly support the main claims.

**Q2(5) Reproducibility:**

3: Good: Key resources (e.g., proofs, code, data) are available and key details (e.g., proofs, experimental setup) are sufficiently well-described for competent researchers to confidently reproduce the main results.

**Q3 Main Strengths:**

1) The motivation of the proposed approach is clear: the benefits of VDE (adaptation to the local geometry of the data) and its challenges (computation and the identification of submanifolds where the data is concentrated) are well described.
2) The proposed approach is sound and addresses the challenges discussed in the introduction
3) The experimental evaluation addresses various aspects of the proposed method, and gives a qualitative idea about the performance, the dependency of parameters, and the accuracy of the result.

**Q4 Main Weakness:**

1) The paper indicates that the proposed method provides solutions for the stated challenges but only in a constrained way. For example, for the MNIST experiments, the dimension of the data had to be reduced by PCA to 10. I assume that ideally, when the method is really able to identify the manifolds where the data concentrates, such a step would not be necessary.
2) The synthetic data in the experiments is always Gaussian. I expect this to work well with the employed Gaussian kernel. Interesting would be to see what happens when other data distributions are used. The experiments generally are fine, but they could be extended with other competitors and more datasets.

**Q5 Detailed Comments To The Authors:**

* One of the motivations for compactification is the identification of submanifolds of the data. It's unclear to me how this is achieved with the Gaussian kernel, which does not identify/reflect submanifolds of the data.
* The notation for the functions $C(x)$ and $C(p)$ is overloaded and not easy to distinguish. Better use varying function names.
* The motivation against using a bounded region before Fig. 3 is unclear to me. The data we have does usually not extend infinitely into the direction of a submanifold and hence I would say that the assumption of a bounded region is sensible to me.
* How strong is the assumption of Thm 4.1 that $\rho$ has support on the whole $\mathbb{R}^n$? Doesn't that contradict the submanifold hypothesis?
* The definition of Dirac's delta in Prop. 4.2 is unclear to me. When does it return 1?
* I was missing an interpretation of Figure 6, the number of versors doesn't seem to have a big impact in comparison to the initialization. What can we learn from that?

### After Rebuttal Thoughts
The authors wrote a nice rebuttal. I get that the local approximations given by the Gaussian kernel are more flexible to adapt to the data manifold with a small noise sleeve. I was thinking that it would be probably better to learn the covariance matrix of the Gaussians too, such that also locally low-dimensionality is reflected. This might then be something for further work.

The only thing that still seems weird is the practical requirement to use PCA beforehand. PCA is a linear dimensionality reduction method and I believe that reducing the data to 10PCs might easily fail to capture the original manifold of the data.
With regard to the notation of $C(.)$ I would just adapt the Definition 2.1 such that $C(.)$ returns the cell of the closest centroid, then that would be clear.

I leave my score of weak accept but I'll note that it's with a tendency towards accept.


**Q7 Justification For Your Score:**

Good paper that provides some interesting directions for density approximation techniques based on a Voronoi tesselation. The posed challenges are not really solved, some of the claims seem to be a bit misleading, and the experimental evaluation could be strengthened. Yet, all in all, the paper is sound and has a contribution that could inspire future work.

**Q9 Complying With Reviewing Instructions:**

1: Yes.

---

### Decision · Program_Chairs · 2022-05-15

**Decision:**

Accept (Poster)

**Comment:**

Meta Review: All the reviewers recommended acceptance, so I'm happy to accept this paper.